# Model-Predictive Policy Learning with Uncertainty Regularization for Driving in Dense Traffic

**Mikael Henaff** [*]
Courant Institute, New York University
Microsoft Research, NYC
`mbh305@nyu.edu`

**Alfredo Canziani** [*]
Courant Institute, New York University
`canziani@nyu.edu`

**Yann LeCun**
Courant Institute, New York University
Facebook AI Research
`yann@cs.nyu.edu`

## Abstract

Learning a policy using only observational data is challenging because the distribution of states it induces at execution time may differ from the distribution observed during training. We propose to train a policy by unrolling a learned model of the environment dynamics over multiple time steps while explicitly penalizing two costs: the original cost the policy seeks to optimize, and an uncertainty cost which represents its divergence from the states it is trained on. We measure this second cost by using the uncertainty of the dynamics model about its own predictions, using recent ideas from uncertainty estimation for deep networks. We evaluate our approach using a large-scale observational dataset of driving behavior recorded from traffic cameras, and show that we are able to learn effective driving policies from purely observational data, with no environment interaction.

## 1 Introduction

In recent years, model-free reinforcement learning methods using deep neural network controllers have proven effective on a wide range of tasks, from playing video or text-based games (Mnih et al., 2015; 2016; Narasimhan et al., 2015) to learning algorithms (Zaremba et al., 2015) and complex locomotion tasks (Lillicrap et al., 2015; Zhang et al., 2015). However, these methods often require a large number of interactions with the environment in order to learn. While this is not a problem if the environment is simulated, it can limit the application of these methods in realistic environments where interactions with the environment are slow, expensive or potentially dangerous. Building a simulator where the agent can safely try out policies without facing real consequences can mitigate this problem, but requires human engineering effort which increases with the complexity of the environment being modeled.

Model-based reinforcement learning approaches try to learn a model of the environment dynamics, and then use this model to plan actions or train a parameterized policy. A common setting is where an agent alternates between collecting experience by executing actions using its current policy or dynamics model, and then using these experiences to improve its dynamics model. This approach has been shown empirically to significantly reduce the required number of environment interactions needed to obtain an effective policy or planner (Atkeson & Santamaria, 1997; Deisenroth & Rasmussen, 2011; Nagabandi et al., 2017; Chua et al., 2018).

Despite these improvements in sample complexity, there exist settings where even a *single* poor action executed by an agent in a real environment can have consequences which

---

[*]Equal contribution.

are not acceptable. At the same time, with data collection becoming increasingly inexpensive, there are many settings where observational data of an environment is abundant. This suggests a need for algorithms which can learn policies primarily from observational data, which can then perform well in a real environment. Autonomous driving is an example of such a setting: on one hand, trajectories of human drivers can be easily collected using traffic cameras (Halkias & Colyar, 2006), resulting in an abundance of observational data; on the other hand, learning through interaction with the real environment is not a viable solution.

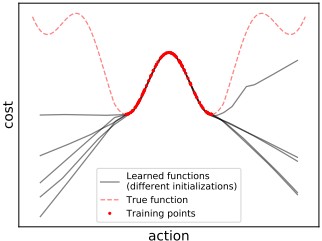

However, learning policies from purely observational data is challenging because the data may only cover a small region of the space over which it is defined. If the observational data consists of state-action pairs produced by an expert, one option is to use imitation learning (Pomerleau, 1991). However, this is well-known to suffer from a mismatch between the states seen at training and execution time (Ross & Bagnell, 2010). Another option is to learn a dynamics model from observational data, and then use it to train a policy (Nguyen & Widrow, 1989). However, the dynamics model may make arbitrary predictions outside the domain it was trained on,

Figure 1: Different models fitted on training points which cover a limited region the function's domain. Models make arbitrary predictions outside of this region.

which may wrongly be associated with low cost (or high reward) as shown in Figure 1. The policy network may then exploit these errors in the dynamics model and produce actions which lead to wrongly optimistic states. In the interactive setting, this problem is naturally self-correcting, since states where the model predictions are wrongly optimistic will be more likely to be experienced, and thus will correct the dynamics model. However, the problem persists if the dataset of environment interactions which the model is trained on is fixed.

In this work, we propose to train a policy while explicitly penalizing the mismatch between the distribution of trajectories it induces and the one reflected in the training data. We use a learned dynamics model which is unrolled for multiple time steps, and train a policy network to minimize a differentiable cost over this rolled-out trajectory. This cost contains two terms: a policy cost which represents the objective the policy seeks to optimize, and an uncertainty cost which represents its divergence from the states it is trained on. We measure this second cost by using the uncertainty of the dynamics model about its own predictions, calculated using dropout. We apply our approach in the context of learning policies to drive an autonomous car in dense traffic, using a large-scale dataset of real-world driving trajectories which we also adapt into an environment for testing learned policies [1]. We show that model-based control using this additional uncertainty regularizer substantially outperforms unregularized control, and enables learning good driving policies using *only* observational data with no environment interaction or additional labeling by an expert. We also show how to effectively leverage an action-conditional stochastic forward model using a modified posterior distribution, which encourages the model to maintain sensitivity to input actions.

## 2 Model-Predictive Policy Learning with Uncertainty Regularization

We assume we are given a dataset of observational data which consists of state-action pairs $\mathcal{D} = \{(s_t, a_t)\}_t$. We first describe our general approach, which consists of two steps: learning an action-conditional dynamics model using the collected observational data, and then using this model to train a fast, feedforward policy network which minimizes both a policy cost and an uncertainty cost.

### 2.1 Action-conditional Forward Model

In this work, we consider recent approaches for stochastic prediction based on Variational Autoencoders (Kingma & Welling, 2013; Babaeizadeh et al., 2017; Denton & Fergus, 2018). The stochastic model $f_\theta(s_{1:t}, a_t, z_t)$ takes as input a sequence of observed or previously predicted states $s_{1:t}$, an action $a_t$, and a latent variable $z_t$ which represents the information about the next state $s_{t+1}$ which

---

[1]Dataset and environment can be found at `https://github.com/Atcold/pytorch-PPUU`

is not a deterministic function of the input. During training, latent variables are sampled from a distribution whose parameters are output by a posterior network $q_\phi(s_{1:t}, s_{t+1})$ conditioned on the past inputs and true targets. This network is trained jointly with the rest of the model using the reparameterization trick, and a term is included in the loss to minimize the KL divergence between the posterior distribution and a fixed prior $p(z)$, which in our case is an isotropic Gaussian.

The per-sample loss used for training the stochastic model is given by:

$$\mathcal{L}(\theta, \phi; s_{1:t}, s_{t+1}, a_t) = \|s_{t+1} - f_\theta(s_{1:t}, a_t, z_t)\|_2^2 + \beta D_{KL}(q_\phi(z|s_{1:t}, s_{t+1})\|p(z)) \qquad (1)$$

After training, different future predictions for a given sequence of frames can be generated by sampling different latent variables from the prior distribution.

Recent models for stochastic video prediction (Babaeizadeh et al., 2017; Denton & Fergus, 2018) do not use their model for planning or training a policy network, and parameterize the posterior distribution over latent variables using a diagonal Gaussian. In our case, we are training an action-conditional video prediction model which we will later use to train a policy. This leads to an additional requirement: it is important for the prediction model to accurately respond to input actions, and not use the latent variables to encode factors of variation in the outputs which are due to the actions. To this end we propose to use a mixture of two Gaussians, with one component fixed to the prior, as our posterior distribution:

$$
\begin{aligned}
(\mu_\phi, \sigma_\phi) &= q_\phi(s_{1:t}, s_{t+1}) \\
u &\sim \mathcal{B}(p_u) \\
z_t &\sim (1 - u) \cdot \mathcal{N}(\mu_\phi, \sigma_\phi) + u \cdot p(z)
\end{aligned}
$$

This can be seen as applying a form of global dropout to the latent variables at training time [2], and forces the prediction model to extract as much information as possible from the input states and actions by making the latent variable independent of the output with some probability. In our experiments we will refer to this parameterization as $z$-dropout.

## 2.2 Training a Policy Network with Uncertainty Minimization

Once the forward model is trained, we use it to train a parameterized policy network $\pi_\psi$, which we assume to be stochastic. We first sample an initial state sequence $s_{1:t}$ from the training set, unroll the forward model over $T$ time steps, and backpropagate gradients of a differentiable objective function with respect to the parameters of the policy network (shown in Figure 2). During this process the weights of the forward model are fixed, and only the weights of the policy network are optimized. This objective function contains two terms: a *policy cost* $C$, which reflects the underlying objective the policy is trying to learn, and an *uncertainty cost* $U$, which reflects how close the predicted state induced by the policy network is to the manifold which the data $\mathcal{D}$ is drawn from.

Training the policy using a stochastic forward model involves solving the following problem, where latent variables are sampled from the prior and input into the forward model at every time step:

$$\operatorname*{argmin}_\psi \left[ \sum_{i=1}^{T} C(\hat{s}_{t+i}) + \lambda U(\hat{s}_{t+i}) \right], \text{ such that: } \begin{cases} z_{t+i} \sim p(z) \\ \hat{a}_{t+i} \sim \pi_\psi(\hat{s}_{t+i-1}) \\ \hat{s}_{t+i} = f(\hat{s}_{t+i-1}, \hat{a}_{t+i}, z_{t+i}) \end{cases}$$

The uncertainty cost $U$ is applied to states predicted by the forward model, and could reflect any measure of their likelihood under the distribution the training data is drawn from. We propose here a general form based on the uncertainty of the dynamics model, which is calculated using dropout. Intuitively, if the dynamics model is given a state-action pair from the same distribution as $\mathcal{D}$ (which

---

[2]If the variances of both Gaussians are zero, this becomes equivalent to applying dropout to the latent code where all units are either set to zero or unchanged with probability $p_u$.

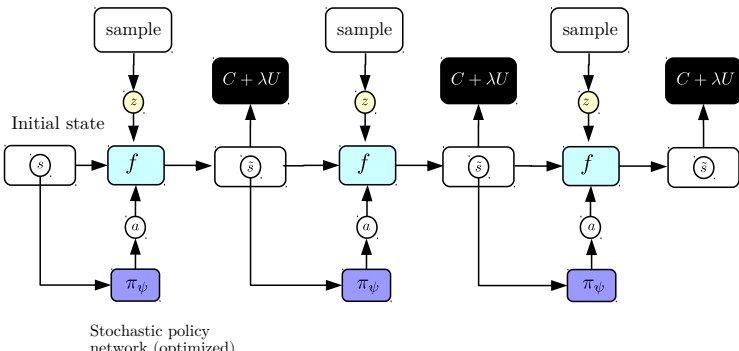

Figure 2: Training the policy network using the stochastic forward model. Gradients with respect to costs associated with predicted states are passed through the unrolled forward model into a policy network.

it was trained on), it will have low uncertainty about its prediction. If it is given a state-action pair which is outside this distribution, it will have high uncertainty.

Dropout (Hinton et al., 2012; Srivastava et al., 2014) is a regularization technique which consists of randomly setting hidden units in a neural network to zero with some probability. The work of (Gal & Ghahramani, 2016) showed that estimates of the neural network's uncertainty for a given input can be obtained by calculating the covariance of its outputs taken over multiple dropout masks. We note that this uncertainty estimate is the composition of differentiable functions: each of the models induced by applying a different dropout mask is differentiable, as is the covariance operator. Furthermore, we can summarize the covariance matrix by taking its trace (which is equal to the sum of its eigenvalues, or equivalently the sum of the variances of the outputs across each dimension), which is also a differentiable operation. This provides a scalar estimate of uncertainty which is differentiable with respect to the input.

More precisely, let $f_{\theta_1}, ..., f_{\theta_K}$ denote our prediction model with $K$ different dropout masks applied to its hidden units (this can also be viewed as changing its weights). We define our scalar measure of uncertainty $U$ as follows:

$$U(\hat{s}_{t+1}) = \text{tr}\Big[\text{Cov}[\{f_{\theta_k}(s_{1:t}, a_t, z_t)\}_{k=1}^K]\Big]$$
$$= \sum_{j=1}^{d} \text{Var}(\{f_{\theta_k}(s_{1:t}, a_t, z_t)_j\}_{k=1}^K)$$

where $d$ is the dimensionality of the output. Minimizing this quantity with respect to actions encourages the policy network to produce actions which, when plugged into the forward model, will produce predictions which the forward model is confident about [3].

A simple way to define $U$ given an initial sequence of states $s_{1:t}$ from $\mathcal{D}$ would be to set $U(\hat{s}_{t+k}) = \|\hat{s}_{t+k} - s_{t+k}\|_2$, which would encourage the policy network to output actions which lead to a similar trajectory as the one observed in the dataset. This leads to a set of states which the model is presumably confident about, but may not be a trajectory which also satisfies the policy cost $C$ unless the dataset $\mathcal{D}$ consists of *expert* trajectories. If this is the case, setting $C(\hat{s}_{t+i}) = U(\hat{s}_{t+i}) = \frac{1}{2}\|\hat{s}_{t+k} - s_{t+k}\|_2$ gives a model-based imitation learning objective which simultaneously optimizes the policy cost and the uncertainty cost. The problem then becomes:

---

[3]In practice, we apply an additional step to normalize this quantity across different modalities and rollout lengths, which is detailed in Appendix D.

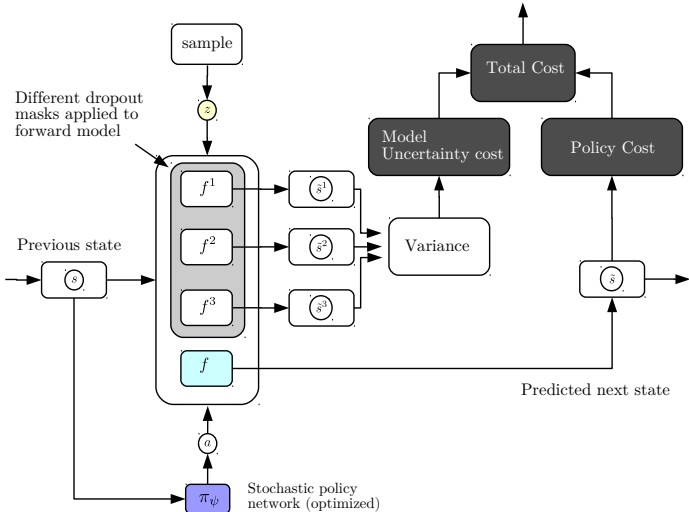

Figure 3: Training the policy network using the differentiable uncertainty cost, calculated using dropout.

$$\underset{\psi}{\text{argmin}} \Big[ \sum_{i=1}^{T} \|\hat{s}_{t+i} - s_{t+i}\|_2 \Big], \text{ such that: } \begin{cases} z_{t+i} \sim q_\phi(z|s_{1:t}, s_{t+1}) \\ \hat{a}_{t+i} \sim \pi_\psi(\hat{s}_{t+i-1}) \\ \hat{s}_{t+i} = f(\hat{s}_{t+i-1}, \hat{a}_{t+i}, z_{t+i}) \end{cases}$$

We call the first approach MPUR , for Model-Predictive Policy with Uncertainty Regularization, and the second MPER , for Model-Predictive Policy with Expert Regularization. A key feature of both approaches is that we optimize the objective over $T$ time steps, which is made possible by our learned dynamics model. This means that the actions will receive gradients from multiple time steps ahead, which will penalize actions which lead to large divergences from the training manifold further into the future, even if they only cause a small divergence at the next time step.

## 2.3 RELATIONSHIP TO BAYESIAN NEURAL NETWORKS

Our MPUR approach can be viewed as training a Bayesian neural network (BNN) (Neal, 1995) with latent variables using variational inference (Jordan et al., 1999; Kingma & Welling, 2013). The distribution over model predictions for $s_{t+1}$ is given by:

$$p(s_{t+1}|s_{1:t}, a, \mathcal{D}) = \int p(s_{t+1}|f_\theta(s_{1:t}, a, z))p(\theta, z|\mathcal{D})d\theta dz$$

The distribution $p(\theta, z|\mathcal{D})$ reflects the posterior over model weights and latent variables given the data, and is intractable to evaluate. We instead approximate it with the variational distribution $q$ parameterized by $\eta = \{\phi, \theta^*\}$:

$$q_\eta(z, \theta) = q_\phi(z|s_{1:t}, s_{t+1}) \cdot q_{\theta^*}(\theta)$$

Here $q_\phi$ represents a distribution over latent variables represented using a posterior network with parameters $\phi$, which could be a diagonal Gaussian or the mixture distribution described in Section 2.1. The distribution $q_{\theta^*}$ is the *dropout approximating distribution* over forward model parameters described in (Gal & Ghahramani, 2016), a mixture of two Gaussians with one mean fixed at zero. We show in Appendix B that training the stochastic forward model with dropout by minimizing the loss function in Equation 1 is approximately minimizing the Kullback-Leibler divergence between this approximate posterior and the true posterior.

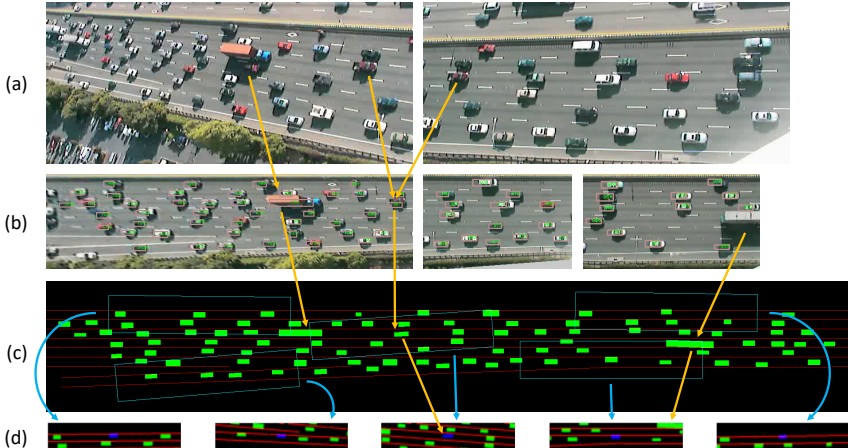

Figure 4: Preprocessing pipeline for the NGSIM-I80 data set. Orange arrows show same vehicles across stages. Blue arrows show corresponding extracted context state. (a) Snapshots from two of the seven cameras. (b) View point transformation, car localisation and tracking. (c) Context states are extracted from rectangular regions surrounding each vehicle. (d) Five examples of context states $i_t$ extracted at the previous stage.

Once the forward model is trained, for a given input we can obtain an approximate distribution over outputs $p(\hat{s}_{t+1}|s_{1:t}, a)$ induced by the approximate posterior by sampling different latent variables and dropout masks. We now show that the covariance of the outputs $\hat{s}_{t+1}$ can be decomposed into a sum of two covariance matrices which represent the aleatoric and epistemic uncertainty, using a similar approach as (Depeweg et al., 2018). Using the conditional covariance formula we can write:

$$\text{cov}(\hat{s}_{t+1}|s_{1:t}, a) = \text{cov}_\theta(\mathbb{E}_z[\hat{s}_{t+1}|s_{1:t}, a, \theta]) + \mathbb{E}_\theta[\text{cov}_z(\hat{s}_{t+1}|s_{1:t}, a, \theta)] \qquad (2)$$

The first term is the covariance of the random vector $\mathbb{E}_z[\hat{s}_{t+1}|s_{1:t}, a, \theta]$ when $\theta \sim q_{\theta^*}(\theta)$. This term ignores any contribution to the variance from $z$ and only considers the effect of $\theta$. As such it represents the epistemic uncertainty. The second term represents the covariance of the predictions obtained by sampling different latent variables $z \sim p(z)$ averaged over different dropout masks, and ignores any contribution to the variance from $\theta$. As such it represents the aleatoric uncertainty. Our uncertainty penalty explicitly penalizes the trace of the first matrix where the expectation over $z$ is approximated by a single sample from the prior. Note also that the covariance matrix corresponding to the aleatoric uncertainty will change depending on the inputs. This allows our approach to handle heteroscedastic environments, where the aleatoric uncertainty will vary for different inputs.

## 3  DATASET AND PLANNING ENVIRONMENT

We apply our approach to learn driving policies using a large-scale dataset of driving videos taken from traffic cameras. The Next Generation Simulation program's Interstate 80 (NGSIM I-80) dataset (Halkias & Colyar, 2006) consists of 45 minutes of recordings from traffic cameras mounted over a stretch of highway. The driver behavior is complex and includes sudden accelerations, lane changes and merges which are difficult to predict; as such the dataset has high environment (or aleatoric) uncertainty. After recording, a viewpoint transformation is applied to rectify the perspective, and vehicles are identified and tracked throughout the video. This yields a total 5596 car trajectories, which we split into training (80%), validation (10%) and testing sets (10%). In all, the dataset contains approximately 2 million transitions.

We then applied additional preprocessing to obtain a state and action representation $(s_t, a_t)$ for each car at each time step, suitable for learning an action-conditional predictive model. Our state representation $s_t$ consists of two components: an image $i_t$ representing the neighborhood of the car, and a vector $u_t$ representing its current position and velocity. The images $i_t$ are centered around

the ego car and encode both the lane emplacements and the locations of other cars. Each image has 3 channels: the first (red) encodes the lane markings, the second (green) encodes the locations of neighboring cars, which are represented as rectangles reflecting the dimensions of each car, and the third channel (blue) represents the ego car, also scaled to the correct dimensions. This is summarized in Figure 4. The action $a_t$ at a given time step consists of a 2-dimensional vector representing the acceleration/braking of the car and its change in steering angle. We also define two cost functions which together make up the policy cost: a proximity cost which reflects how close the ego car is to neighboring cars, and a lane cost which reflects how much the ego car overlaps with lane markings. These are represented as a cost vector at each timestep, $c_t = (C_{\text{proximity}}(s_t), C_{\text{lane}}(s_t))$. Full details can be found in Appendix A.

We also adapted this dataset to be used as an environment to evaluate learned policies, with the same interface as OpenAI Gym (Brockman et al., 2016). Choosing a policy for neighboring cars is challenging due to a cold-start problem: to accurately evaluate a learned policy, the other cars would need to follow human-like policies which would realistically react to the controlled car, which are not available. We take the approach of letting all the other cars in the environment follow their trajectories from the dataset, while a single car is controlled by the policy we seek to evaluate. This approach avoids hand-designing a policy for the neighboring cars which would likely not reflect the diverse nature of human driving. The limitation is that the neighboring cars do not react to the controlled car, which likely makes the problem more difficult as they do not try to avoid collisions.

## 4 RELATED WORK

A number of authors have explored the use of learned, action-conditional forward models which are then used for planning, starting with classic works in the 90's (Nguyen & Widrow, 1990; Schmidhuber, 1990; Jordan & Rumelhart, 1992), and more recently in the context of video games (Oh et al., 2015; Pascanu et al., 2017; Weber et al., 2017), robotics and continous control (Finn et al., 2016; Agrawal et al., 2016; Nagabandi et al., 2017; Srinivas et al., 2018). Our approach to learning policies by backpropagating through a learned forward model is related to the early work of (Nguyen & Widrow, 1989) in the deterministic case, and the SVG framework of (Heess et al., 2015) in the stochastic case. However, neither of these approaches incorporates a term penalizing the uncertainty of the forward model when training the policy network.

The works of (McAllister & Rasmussen, 2016; Chua et al., 2018) also used model uncertainty estimates calculated using dropout in the context of model-based reinforcement learning, but used them for sampling trajectories during the forward prediction step. Namely, they applied different dropout masks to simulate different state trajectories which reflect the distribution over plausible models, which were then averaged to produce a cost estimate used to select an action.

Our model uncertainty penalty is related to the cost used in (Kahn et al., 2017), who used dropout and model ensembling to compute uncertainty estimates for a binary action-conditional collision detector for a flying drone. These estimates were then used to select actions out of a predefined set which yielded a good tradeoff between speed, predicted chance of collision and uncertainty about the prediction. In our work, we apply uncertainty estimates to the predicted high-dimensional states of a forward model at every time step, summarize them into a scalar, and backpropagate gradients through the unrolled forward model to then train a policy network by gradient descent.

The work of (Depeweg et al., 2018) also proposed adding an uncertainty penalty when training paramaterized policies, but did so in the context of BNNs trained using $\alpha$-divergences applied in low-dimensional settings, whereas we use variational autoencoders combined with dropout for high-dimensional video prediction. $\alpha$-BNNs can yield better uncertainty estimates than variational inference-based methods, which can underestimate model uncertainty by fitting to a local mode of the exact posterior (Depeweg et al., 2016; Li & Gal, 2017). However, they also require computing multiple samples from the distribution over model weights when training the forward model, which increases memory requirements and limits scalability to high-dimensional settings such as the ones we consider here.

The problem of covariate shift when executing a policy learned from observational data has been well-recognized in imitation learning (Pomerleau, 1991; Ross & Bagnell, 2010). The work of (Ross et al., 2011) proposed a method to efficiently use expert feedback (if available) to correct this shift,

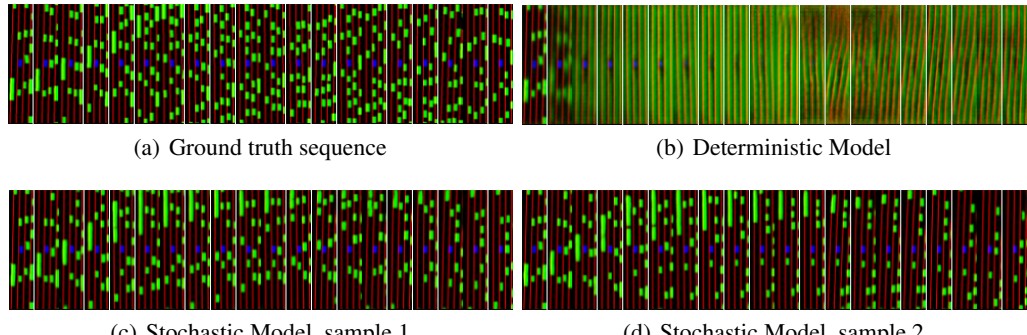

(a) Ground truth sequence

(b) Deterministic Model

(c) Stochastic Model, sample 1

(d) Stochastic Model, sample 2

Figure 5: Video prediction results using a deterministic and stochastic model over 200 time steps (images are subsampled across time). Two different future predictions are generated by the stochastic model by sampling two different sequences of latent variables. The deterministic model averages over possible futures, producing blurred predictions.

which has also been applied in the context of autonomous driving (Zhang & Cho, 2016). Our approach also addresses covariate shift, but does so without querying an expert.

Our MPER approach is related to the work of (Englert et al., 2013), who also performed imitation learning at the level of trajectories rather than individual actions. They did so in low-dimensional settings using Gaussian Processes, whereas our method uses an unrolled neural network representing the environment dynamics which can be applied to high-dimensional state representations. The work of (Baram et al., 2017) also used a neural network dynamics model in the context of imitation learning, but did so in the interactive setting to minimize a loss produced by a discriminator network.

Several works have used deep learning models for autonomous driving, either to learn policies through imitation learning (Pomerleau, 1991; LeCun et al., 2006; Bojarski et al., 2016; Pan et al., 2017) or for modeling vehicle dynamics (Williams et al., 2017). These works focused on lane following or avoiding static obstacles in visually rich environments and did not consider settings with dense moving traffic. The work of (Sadigh et al., 2016) developed a model of the interactions between the two drivers which was then used to plan actions in simple settings, using symbolic state representations. In our work, we consider the problem of learning driving policies in dense traffic, using high-dimensional state representations which reflect the neighborhood of the ego car.

## 5 EXPERIMENTS

We now report experimental results. We designed a deterministic and stochastic forward model to model the state and action representations described in Section 3, using convolutional layers to process the images $i_t$ and fully-connected layers to process the vectors $u_t$ and actions $a_t$. All model details can be found in Appendix C and training details can be found in Appendix D. Code and additional video results for the model predictions and learned policies can be found at the following URL: `https://sites.google.com/view/model-predictive-driving/home`.

### 5.1 PREDICTION RESULTS

We first generated predictions using both deterministic and stochastic forward models, shown in Figure 5. The deterministic model produces predictions which become increasingly blurry, while the stochastic model produces predictions which stay sharp far into the future. By sampling different sequences of latent variables, different future scenarios are generated. Note that the two sequences generated by the stochastic model are different from the ground truth future which occurs in the dataset. This is normal as the future observed in the dataset is only one of many possible ones. Additional video generations can be viewed at the URL.

| Method | Model | Mean Distance | Success Rate (%) |
|---|---|---|---|
| Human | | 209.4 | 100.0 |
| No action | | 87.3 | 16.2 |
| 1-step IL | | $73.8 \pm 7.9$ | $7.3 \pm 4.1$ |
| SVG | stochastic | $17.1 \pm 4.3$ | $0.0 \pm 0.0$ |
| VG | deterministic | $11.9 \pm 4.2$ | $0.0 \pm 0.0$ |
| MPUR | stochastic+$z$-dropout | $171.2 \pm 4.5$ | $74.8 \pm 3.0$ |
| MPUR | stochastic | $166.8 \pm 2.4$ | $71.8 \pm 1.0$ |
| MPUR | deterministic | $162.4 \pm 2.8$ | $69.1 \pm 1.6$ |
| MPER | stochastic | $70.0 \pm 8.0$ | $4.6 \pm 2.1$ |
| MPER | deterministic | $157.4 \pm 0.7$ | $63.7 \pm 0.5$ |

(a)

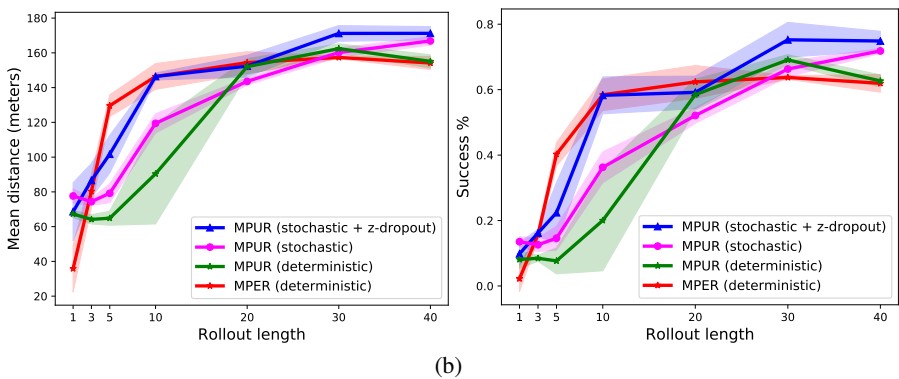

(b)

| Human | The human trajectories observed in the testing set, which are all collision-free. |
|---|---|
| No action | A policy which outputs an action of zero, maintaining constant speed and direction. |
| 1-step IL | A policy network trained with single-step imitation learning. |
| SVG | A policy network trained with stochastic value gradients. This is the same setup as (Heess et al., 2015), with the difference that the agent does not interact with the environment and learns from a fixed observational dataset. |
| VG | A policy trained with value gradients, using the deterministic forward model. This is similar to SVG, but does not involve latent variables. |
| MPUR | A policy trained with MPUR , using a deterministic or stochastic model. A cost term is included to penalize the uncertainty of the dynamics model. |
| MPER | A policy trained with MPER , using a deterministic or stochastic model. The policy is trained to match expert trajectories from the training set. |

(c)

Figure 6: a) Performance different methods, measured in success rate and distance travelled. Including a cost term penalizing the dynamics model's uncertainty is essential for good performance. Using the modified posterior distribution ($z$-dropout) improves performance when using the stochastic forward model. b) Training policies by performing longer rollouts through the environment dynamics model also significantly improves performance. c) Summary of compared methods.

## 5.2 POLICY EVALUATION RESULTS

We evaluated policies using two measures: whether the controlled car reaches the end of the road segment without colliding into another car or driving off the road, and the distance travelled before the episode ends. Policies which collide quickly will travel shorter distances.

We compared our approach against several baselines which can also learn from observational data, which are described in Figure 6c. Table 6a compares performance for the different methods, optimized over different rollout lengths. The 1-step imitation learner, SVG and VG all perform poorly, and do not beat the simple baseline of performing no action. Both MPUR and MPER significantly outperform the other methods. Videos of the learned policies for both MPER and MPUR driving in

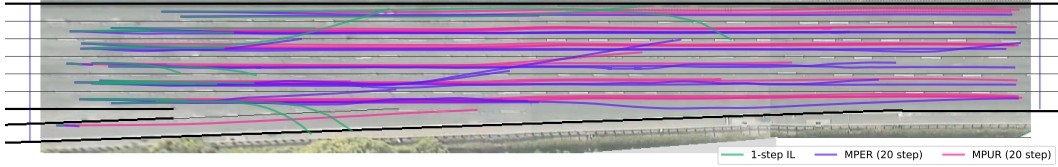

Figure 7: Trajectories for MPUR , MPER and a single-step imitation learner policies over the stretch of highway. The single-step imitation learner policy often veers across lanes, whereas the other policies stay within. These policies also change their speed to avoid collisions (see URL).

| Method | Mean Distance (m) | Success Rate (%) | Total Predicted Cost | $U$ |
|--------|-------------------|------------------|----------------------|-----|
| VG | 14.9 | 0.0 | 0.05 | 1081.2 |
| MPUR | 168.8 | 73.5 | 0.13 | 0.4 |

Table 1: Policy and uncertainty costs with and without uncertainty regularization. The policy trained with unregularized VG exploits errors in the forward model to produce actions which yield low predicted cost but high uncertainty. Including the uncertainty cost yields higher predicted cost, but better performance when the policy is executed in the environment.

the environment can be found at the URL. The policies learn effective behaviors such as braking, accelerating and turning to avoid other cars. Figure 7 shows trajectories on the map for different methods. We see that the single-step imitation learner produces divergent trajectories which turn into other lanes, whereas the MPUR and MPER methods show trajectories which primarily stay within their lanes.

MPUR becomes equivalent to VG in the deterministic setting if we remove the uncertainty penalty, and the large difference in performance shows that including this penalty is essential. Table 1 shows the average predicted policy cost and uncertainty cost of the two methods. VG produces much lower predicted policy cost, yet very high uncertainty cost. This indicates that the actions the policy produces induce a distribution over states which the forward model is highly uncertain about. The policy trained with MPUR produces higher policy cost estimates, but lower uncertainty cost, and performs much better when executed in the environment.

The stochastic model trained with a standard Gaussian posterior yields limited improvement over the deterministic model. However, the stochastic model trained with the $z$-dropout parameterization yields a significant improvement. Comparisons of action-conditional predictions using both models can be seen at the URL. The standard model is less responsive to input actions than the model trained with $z$-dropout, which likely accounts for their difference in performance. We hypothesize that the standard model uses its latent variables to encode factors of variation in the output which are in fact due to the actions. Using $z$-dropout discourages this since information in the output cannot be encoded in the latent variables the times they are sampled from the prior, and the loss can better be lowered by predicting the outputs from the actions instead. Please see Appendix E for additional experiments and discussion of this phenomenon.

Figure 6b shows the performance of MPUR and MPER for different rollout lengths. All methods see their performance improve dramatically as we increase the rollout length, which encourages the distribution of states the policy induces and the training distribution to match over longer time horizons. We also see that the stochastic model with $z$-dropout outperforms the standard stochastic model as well as the deterministic model over most rollout lengths.

## 6 CONCLUSION

In this work, we proposed a general approach for learning policies from purely observational data. The key elements are: i) a learned stochastic dynamics model, which is used to optimize a policy cost over multiple time steps, ii) an uncertainty term which penalizes the divergence of the trajectories induced by the policy from the manifold it was trained on, and iii) a modified posterior distribution

which keeps the stochastic model responsive to input actions. We have applied this approach to a large observational dataset of real-world traffic recordings, and shown it can effectively learn policies for navigating in dense traffic, which outperform other approaches which learn from observational data. However, there is still a sizeable gap between the performance of our learned policies and human performance. We release both our dataset and environment, and encourage further research in this area to help narrow this gap. We also believe this provides a useful setting for evaluating generative models in terms of their ability to produce good policies. Finally, our approach is general and could potentially be applied to many other settings where interactions with the environment are expensive or unfeasible, but observational data is plentiful.

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

## A    DATASET AND PLANNING ENVIRONMENT

To begin with, we describe the details and preparation of the dataset and planning environment which we used, which are summarized in Figure 4. The Next Generation Simulation program's Interstate 80 (NGSIM I-80) dataset (Halkias & Colyar, 2006) consists of 45 minutes of recordings made of a stretch of highway in the San Francisco Bay Area by cameras mounted on a 30-story building overlooking the highway. The recorded area includes six freeway lanes (including a high-occupancy vehicle lane) and an onramp. The driver behavior is complex and includes sudden accelerations, lane changes and merges which are difficult to predict; as such the dataset has high aleatoric uncertainty. There are three time segments, each of 15 minutes, taken at different times of day which capture the transition between uncongested and congested peak period conditions. After recording, a viewpoint transformation is applied to rectify the perspective, and vehicles are identified and tracked throughout the video; additionally, their size is inferred. This yields a total 5596 car trajectories, represented as sequences of coordinates $\{x_t, y_t\}$. We split these trajectories into training (80%), validation (10%) and testing sets (10%).

We then applied additional preprocessing to obtain suitable representations for learning a predictive model. Specifically, we extracted the following: i) a state representation for each car at each time step $s_t$, which encodes the necessary information to choose an action to take, ii) an action $a_t$ which represents the action of the driver, and iii) a cost $c_t$, which associates a quality measure to each state. We describe each of these below.

**State representation**: Our state representation consists of two components: an image representing the neighborhood of the car, and a vector representing its current position and velocity. For the images, we rendered images centered around each car which encoded both the lane emplacements and the locations of other cars. Each image has 3 channels: the first (red) encodes the lane markings, the second (green) encodes the locations of neighboring cars, which are represented as rectangles reflecting the dimensions of each car, and the third channel (blue) represents the ego car, also scaled to the correct dimensions. All images have dimensions $3 \times 117 \times 24$, and are denoted by $i_t$. [4] Two examples are shown in Figure 8. We also computed vectors $u_t = (p_t, \Delta p_t)$, where $p_t = (x_t, y_t)$ is the position at time $t$ and $\Delta p_t = (x_{t+1} - x_t, y_{t+1} - y_t)$ is the velocity.

---

[4]Another possibility would have been to construct feature vectors directly containing the exact coordinates of neighboring cars, however this presents several difficulties. First, cars can enter and exit the neighborhood, and so the feature vector representing the neighboring cars would either have to be dynamically resized or padded with placeholder values. Second, this representation would not be permutation-invariant, and it is unclear where to place a new car entering the frame. Third, encoding the lane information in vector form would require a parametric representation of the lanes, which is more complicated. Using images representations naturally avoids all of these difficulties.

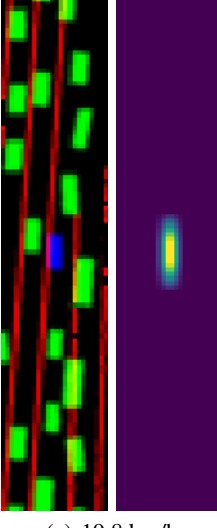 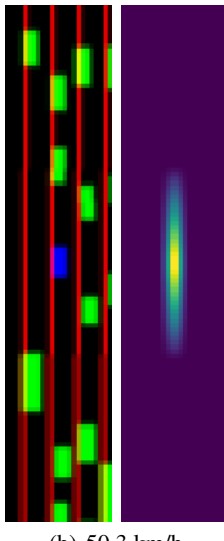

(a) 19.8 km/h                    (b) 50.3 km/h

Figure 8: Image state representations and proximity cost masks for cars going at different speeds. The higher the speed, the longer the safety distance required to maintain low cost.

**Action representation**: Each action vector $a_t$ consists of two components: an acceleration (which can be positive or negative) which reflects the change in speed, and a change in angle. The acceleration at a given time step is computed by taking the difference between two consecutive speeds, while the change in angle is computed by projecting the change in speed along its orthogonal direction:

$$\Delta\text{speed} = \|\Delta p_{t+1}\|_2 - \|\Delta p_t\|_2$$
$$\Delta\text{angle} = (\Delta p_{t+1} - \Delta p_t)^\top (\Delta p_t)_\perp / \|\Delta p_t\|_2$$
$$a_t = (\Delta\text{speed}, \Delta\text{angle})$$

**Cost**: Our cost function has two terms: a proximity cost and a lane cost. The proximity cost reflects how close the ego car is to neighboring cars, and is computed using a mask in pixel space whose width is equal to the width of a lane and whose height depends on the speed of the car. Two examples are shown in Figure 8. This mask is pointwise multiplied with the green channel, and the maximum value is taken to produce a scalar cost. The lane cost uses a similar mask fixed to the size of the car, and is similarly multiplied with the red channel, thus measuring the car's overlap with the lane. Both of these operations are differentiable so that we can backpropagate gradients with respect to these costs through images predicted by a forward model.

This preprocessing yields a set of state-action pairs $(s_t, a_t)$ (with $s_t = (i_t, u_t)$) for each car, which constitute the dataset we used for training our prediction model. We then use the cost function to optimize action sequences at planning time, using different methods which we describe in Section 2.2.

We now describe how we adapted this dataset to be used as an environment to evaluate planning methods. Building an environment for evaluating policies for autonomous driving is not obvious as it suffers from a cold-start problem. Precisely measuring the performance of a given driving policy would require it to be evaluated in an environment where all other cars follow policies which accurately reflect human behavior. This involves reacting appropriately both to other cars in the environment as well as the car being controlled by the policy being evaluated. However, constructing such an environment is not possible as it would require us to already have access to a policy which drives as humans do, which in some sense is our goal in the first place. One could hand-code a driving policy to control the other cars in the environment, however is it not clear how to do so in a way which accurately reflects the diverse and often unpredictable nature of human driving.

```
observation = env.reset()
while not done:
    action = policy(observation)
    observation, reward, done, info = env.step(action)
    env.render()
```

Figure 9: NGSIM planning environment.

We adopt a different approach where we let all other cars in the environment follow their trajectories in the dataset, while controlling one car with the policy we seek to evaluate. The trajectory of the controlled car is updated as a function of the actions output by the policy, while the trajectories of the other cars remain fixed. If the controlled car collides with another car, this is recorded and the episode ends. This approach has the advantage that all other cars in the environment maintain behavior which is close to human-like. The one difference with true human behavior is that the other cars do not react to the car being controlled or try to avoid it, which may cause crashes which would not occur in real life. The driving task is thus possibly made more challenging than in a true environment, which we believe is preferable to using a hand-coded policy. The interface is set up the same way as environments in OpenAI Gym (Brockman et al., 2016), and can be accessed with a few lines of Python code, as shown in Figure 9.

## B  BAYESIAN NEURAL NETWORK FORMULATION DETAILS

Our MPUR approach can be viewed as training a Bayesian neural network (BNN) (Neal, 1995) with latent variables using variational inference (Jordan et al., 1999; Kingma & Welling, 2013). The distribution over model predictions for $s_{t+1}$ is given by:

$$p(s_{t+1}|s_{1:t}, a, \mathcal{D}) = \int p(s_{t+1}|f_\theta(s_{1:t}, a, z))p(\theta, z|\mathcal{D})d\theta dz$$

The distribution $p(\theta, z|\mathcal{D})$ reflects the posterior over model weights and latent variables given the data, and is intractable to evaluate. We instead approximate it with the variational distribution $q$ parameterized by $\eta$:

$$q_\eta(z, \theta) = q_\phi(z|s_{1:t}, s_{t+1}) \cdot q_{\theta*}(\theta)$$

Here $q_\phi$ represents a distribution over latent variables parameterized by $\phi$, which could be a diagonal Gaussian or the mixture distribution described in Section 2.1. The distribution $q_{\theta*}$ is the *dropout approximating distribution* over model parameters described in (Gal & Ghahramani, 2016) (Section 3.2 of Supplement). This defines a mixture of two Gaussians with small variances over each row of each weight matrix in the forward model, with the mean of one Gaussian fixed at zero. The parameters of this distribution are the model weights, and samples can be drawn by applying different dropout masks. The parameters of the variational distribution are thus $\eta = \{\theta^*, \phi\}$, and can be optimized by maximizing the evidence lower bound, which is equivalent to minimizing the Kullback-Leibler divergence between the approximate posterior and true posterior:

$$\mathcal{L}_{ELBO}(\theta^*, \phi; s_{1:t}, s_{t+1}, a) = \int \log p(s_{t+1}|f_\theta(s_{1:t}, a, z))q_\eta(\theta, z)d\theta dz - D_{KL}(q_\eta(\theta, z)||p_0(\theta, z))$$

(3)

Here $p_0(z, \theta) = p_0(z) \cdot p_0(\theta)$ represents a prior over latent variables and model parameters. By applying the chain rule for KL divergences together with the fact that $z$ and $\theta$ are independent, we obtain:

$$D_{KL}(q_\eta(\theta, z)||p_0(\theta, z)) = D_{KL}(q_\phi(z|s_{1:t}, s_{t+1}) \cdot q_{\theta^*}(\theta)||p_0(z) \cdot p_0(\theta))$$
$$= D_{KL}(q_\phi(z|s_{1:t}, s_{t+1})||p_0(z)) + D_{KL}(q_{\theta^*}(\theta|z)||p_0(\theta|z))$$
$$= D_{KL}(q_\phi(z|s_{1:t}, s_{t+1})||p_0(z)) + D_{KL}(q_{\theta^*}(\theta)||p_0(\theta))$$

Setting both Gaussians in $p_0(\theta)$ to have zero mean, the second KL term becomes equivalent to scaled $\ell_2$ regularization on the model parameters, and can be set arbitrarily small (Section 4.2 in Supplement of (Gal & Ghahramani, 2016)). Ignoring this term and approximating the integral in Equation 3 using a single sample, we obtain:

$$\mathcal{L}_{ELBO}(\theta^*, \phi; s_{1:t}, s_{t+1}, a) \approx \log p(s_{t+1}|f_{\bar\theta}(s_{1:t}, a, z)) - D_{KL}(q_\phi(z|s_{1:t}, s_{t+1})||p_0(z))$$

where $\bar\theta \sim q_{\theta^*}(\theta)$ and $z \sim q_\phi(s_{1:t}, s_{t+1})$). Assuming a diagonal Gaussian likelihood on the outputs with constant variance $1/\beta$, we can rewrite this as:

$$\mathcal{L}_{ELBO}(\theta^*, \phi; s_{1:t}, s_{t+1}, a) \approx -\frac{1}{\beta} \cdot \|s_{t+1} - f_{\bar\theta}(s_{1:t}, a, z)\| - D_{KL}(q_\phi(z|s_{1:t}, s_{t+1})||p_0(z))$$

Multiplying by $\beta$ does not change the maximum. We now see that maximizing this quantity is equivalent to minimizing our loss term in Equation 1, i.e. training a variational autoencoder with dropout.

## C  MODEL DETAILS

The architecture of our forward model consists of four neural networks: a state encoder $f_{\text{enc}}$, an action encoder $f_{\text{act}}$, a decoder $f_{\text{dec}}$, and the posterior network $f_\phi$. At every time step, the state encoder takes as input the concatenation of 20 previous states, each of which consists of a context image $i_t$ and a 4-dimensional vector $u_t$ encoding the car's position and velocity. The images $i_{t-20}, ..., i_t$ are run through a 3-layer convolutional network with 64-128-256 feature maps, and the vectors $u_{t-20}, ..., u_t$ are run through a 2-layer fully connected network with 256 hidden units, whose final layers contain the same number of hidden units as the number of elements in the output of the convolutional network (we will call this number $n_H$). The posterior network takes the same input as the encoder network, as well as the the ground truth state $s_{t+1}$, and maps them to a distribution over latent variables, from which one sample $z_t$ is drawn. This is then passed through an expansion layer which maps it to a representation of size $n_H$. The action encoder, which is a 2-layer fully-connected network, takes as input a 2-dimensional action $a_t$ encoding the car's acceleration and change in steering angle, and also maps it to a representation of size $n_H$. The representations of the input states, latent variable, and action, which are all now the same size, are combined via addition. The result is then run through a deconvolutional network with 256-128-64 feature maps, which produces a prediction for the next image $i_{t+1}$, and a 2-layer fully-connected network (with 256 hidden units) which produces a prediction for the next state vector $u_{t+1}$. These are illustrated in Figure C.

The specific updates of the stochastic forward model are given by:

$$(\mu_\phi, \sigma_\phi) \qquad = q_\phi(s_{1:t}, s_{t+1}) \tag{4}$$
$$\epsilon \qquad \sim \mathcal{N}(0, I) \tag{5}$$
$$z_t \qquad = \mu_\phi + \sigma_\phi \cdot \epsilon \tag{6}$$
$$\hat{s}_{t+1} = (\tilde{i}_{t+1}, \tilde{u}_{t+1}) = f_\theta(s_{1:t}, a_t, z_t) \tag{7}$$

The per-sample loss is given by:

$$\ell(s_{1:t}, s_{t+1}) = \|\tilde{i}_t - i_t\|_2^2 + \|\tilde{u}_t - u_t\|_2^2 + \beta D_{KL}(\mathcal{N}(\mu_\phi, \sigma_\phi)||p(z)) \tag{8}$$

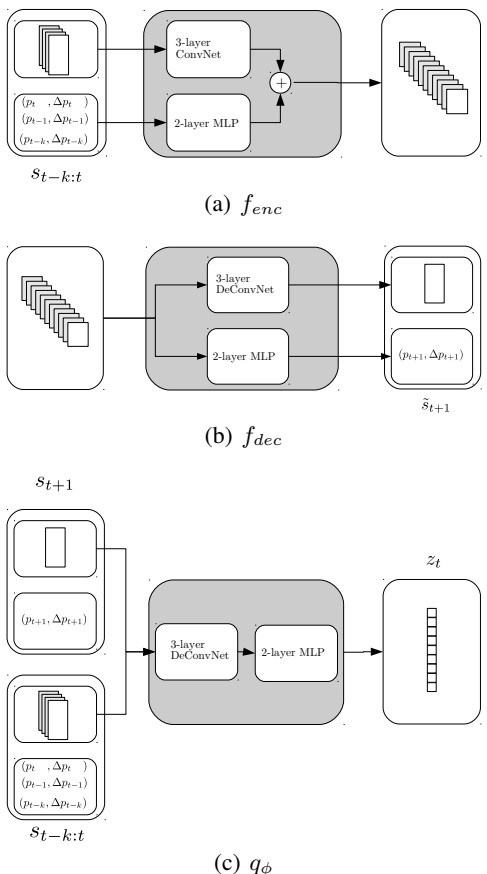

(a) $f_{enc}$

(b) $f_{dec}$

(c) $q_\phi$

Figure 10: Individual components of the prediction model.

We also train a cost predictor which takes as input the states predicted by the forward model and produces a two-dimensional output (one output for the proximity cost, and one output for the lane cost). This consists of a 3-layer encoder followed by a two-layer fully connected network with sigmoid non-linearities as the end to constrain the values between 0 and 1.

# D   TRAINING DETAILS

## D.1   FORWARD MODEL

We trained our prediction model in deterministic mode ($p = 0$) for 200,000 updates, followed by another 200,000 updates in stochastic mode. We save the model after training in deterministic mode and treat it as a deterministic baseline. Our model was trained using Adam (Kingma & Ba, 2014) with learning rate 0.0001 and minibatches of size 64, unrolled for 20 time steps, and with dropout ($p_{dropout} = 0.1$) at every layer, which was necessary for computing the epistemic uncertainty cost when training the policy network.

## D.2   POLICY MODELS

All cars are initialized at the beginning of the road segment with the initial speed they were driving at in the dataset, and then are controlled by the policy being measured. We only report performance for cars in the testing trajectories, which were not used when training the forward model or policy network.

All policy networks have the same architecture: a 3-layer ConvNet with feature maps of size 64-128-256 (which takes 20 consecutive frames as input), followed by 3 fully-connected layers with 256

hidden units each, with the last layer outputting the parameters of a 2D Gaussian distribution from which the action is sampled. All policy networks are trained with Adam with learning rate 0.0001. The MPER and MPUR policies are trained by backpropagation through the unrolled forward model using the reparamaterization trick (Kingma & Welling, 2013). The single-step imitation learner is trained to directly minimize the negative log-likelihood of the ground truth action in the dataset under the parameters output by the policy network. All MPUR policies use a weighting of $\lambda = 0.5$ for the uncertainty cost. Additionally, we detach gradients of predicted costs coming into the states, to prevent the policy from lowering the predicted cost (which is speed-dependent) by slowing down. We found that not doing this can result in the policy slowing excessively, and then attempting to speed up only when another car gets close. We repeat the policy training with 3 random seeds for each method.

The policy cost which we minimize for VG, SVG and MPUR is given by:

$$C = C_{\text{proximity}} + 0.2 \cdot C_{\text{lane}} \tag{9}$$

where $C_{\text{proximity}}$ and $C_{\text{lane}}$ are the proximity and lane costs described in Section 3. This puts a higher priority on avoiding other cars while still encouraging the policy to stay within the lanes. MPUR additionally minimizes $U$, the model uncertainty cost described in Section 2.2.

When computing the uncertainty cost, to compensate for differences in baseline uncertainty across different rollout lengths, we normalize by the empirical mean and variance for every rollout length $t$ of the forward model over the training set, to obtain $\mu_U^t$ and $\sigma_U^t$. We then define our uncertainty cost as follows:

$$U(\hat{s}_{t+1}) \leftarrow \left[ \frac{U(\hat{s}_{t+1}) - \mu_U^t}{\sigma_U^t} \right]_+ \tag{10}$$

If the uncertainty estimate is lower than the mean uncertainty estimate on the training set for this rollout length, this loss will be zero. These are cases where the model prediction is within normal uncertainty ranges. If the uncertainty estimate is higher, this loss exerts a pull to change the action so that the future state will be predicted with higher confidence by the forward model.

## E   ACTION SENSITIVITY EXPERIMENTS

We found the lack of responsiveness of the stochastic forward model to be especially pronounced when it is given a sequence of latent variables inferred from the current training sequence by the posterior network, instead of a sequence of latent variables sampled from the prior (intuitively, using the inferred sequence corresponds to using the future from the dataset, while using a sampled sequence corresponds to a different future). One reason for this difference in responsiveness may be that in the first case, the latent variables are highly dependent whereas in the second they are independent. If action information is encoded in the latent variables, the effects on the output may partially cancel each other out when the latents are independent. However, when they are highly dependent, together they may explain away the effects of the actions input to the forward model.

The table below shows the performance of MPUR policies learned using inferred and sampled latent variables. We see a large drop in performance when using the inferred latent variables. This is consistent with the videos at the URL, which show that the forward model is less sensitive to actions when the latent variables are sampled from the posterior instead of the prior. Note that the $z$-dropout parameterization reduces this problem somewhat.

| Method | Sampling | Mean Distance | Success Rate |
|---|---|---|---|
| MPUR + $z$-dropout | $z_t \sim p(z)$ | 168.2 | 72.1 |
| MPUR | $z_t \sim p(z)$ | 157.4 | 63.6 |
| MPUR + $z$-dropout | $z_t \sim q_\phi(z\|s_{1:t}, s_{t+1})$ | 126.1 | 30.8 |
| MPUR | $z_t \sim q_\phi(z\|s_{1:t}, s_{t+1})$ | 86.0 | 19.2 |

# F MODEL-FREE RESULTS

Although the goal of this work is to learn policies with no environment interaction, for completeness we also report results of running Proximal Policy Optimization (PPO) (Schulman et al., 2017), a state-of-the-art model-free algorithm which learns through interacting with its environment. We used the OpenAI Baselines implementation (Brockman et al., 2016) with the same policy network architecture as for the other methods, and set the reward to be the negative policy cost defined in equation 9. We measure both final performance and cumulative regret, using the success rate as a reward. Cumulative regret is a measure often used in online learning which represents the difference between the agent's accumulated reward and the accumulated reward which would have been obtained by following the optimal policy, which we take to be human performance here. Specifically, the regret at epoch $M$ is given by:

$$\rho(M) = T\mathbb{E}[R^*] - \mathbb{E}[\sum_{m=1}^{M} \hat{R}_m]$$

where $R^*$ represents the reward obtained by following the optimal policy and $R_m$ is the reward obtained by following the policy at epoch $m$. Unlike final performance, regret also reflects poor decisions made during the learning process.

Results are shown in Figure 11. PPO obtains slightly higher final performance than MPUR, but also incurs higher regret as it executes poor policies in the environment during the early stages of learning. In contrast, MPUR learns through observational data and already has a good policy in place when it begins interacting with the environment.

Figure 11: Performance comparison for MPUR and PPO, in terms of a) cumulative regret and b) final performance

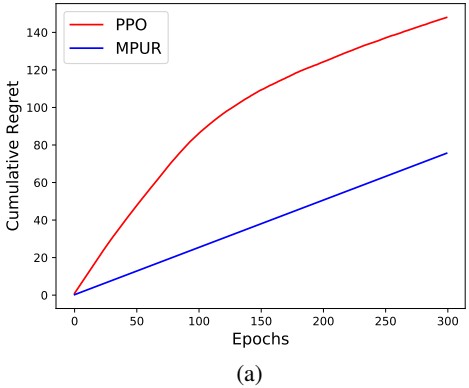

(a)

| Method | Mean Distance | Success Rate |
|---|---|---|
| MPUR + $z$-dropout | $171.2 \pm 4.5$ | $74.8 \pm 3.0$ |
| PPO | $181.0 \pm 6.7$ | $77.8 \pm 5.6$ |

(b)