# OpenReview forum: "Model-Predictive Policy Learning with Uncertainty Regularization for Driving in Dense Traffic"
_ICLR.cc/2019/Conference_

### Official Review · AnonReviewer3 · 2018-10-31
**a good model-based RL attempt for autonomous driving, however, dataset is very limited**

**Rating:** 7
**Confidence:** 5

**Review:**

Pros:
The paper formulates the driving policy problem as a model-based RL problem. Most related work on driving policy has been traditional robotics planning methods such as RRT or model-free RL such as policy gradient methods.

The policy is learned through unrolling a learned model of the environment dynamics over multiple time steps, and training a policy network to minimize a differentiable cost over this rolled-out trajectory.

The cost combine the objective the policy seeks to optimize (proximity to other cars) and an uncertainty cost representing the divergence from the states it is trained on.

Cons:

The model based RL formulation is pretty standard except that the paper has a additional model uncertainty cost.

Realistically, the output of driving policy should be planning decision, i.e. the waypoints instead of steering angles and acceleration / deceleration commands. There does not seem to be a need to solve the control problem using learning since PID and iLQR has solved the control problem very well.

The paper did not seem to reach a conclusion on why stochastic forward model does not yield a clear improvement over the deterministic model. This may be due to the limitation of the dataset or the prediction horizon which seems to be 2 second.

The dataset is only 45 minutes which captured by a camera looking down a small section of the road. So the policies learned might only do lane following and occasionally doing collision avoidance. I would encourage the authors to look into more diverse dataset. See the paper DESIRE: Distant Future Prediction in Dynamic Scenes with Interacting Agents, CVPR 2017.

Overall, the paper makes an interesting contribution: formulate the driving policy problem as a model-based RL problem. The techniques used are pretty standard. There are some insights in the experimental section. However, due to the limitation of the dataset, it is not clear how much the results can generalize to complex settings such as nudging around other cars, cutting in, pedestrian crossing, etc.

Response to rebuttal:
It is good to know that the authors have a new modified VAE posterior distribution for the stochastic model which can achieve significant gain over the deterministic model. Is this empirical and specific to this dataset? Without knowing the details, it is not clear how general this new stochastic model is.

I agree that it is worthwhile to test the model using the 45 minute dataset. However, I still believe the dataset is very limiting and it is not clear how much the experimental results can apply to other large realistic datasets.

My rating stays the same.

---

> ### Author Response · Authors · 2018-11-26
> **Thank you for the review.**
>
> Thank you for the helpful review. We have made updates to the paper, please see our main comment and our answer below.
>
> >”The paper did not seem to reach a conclusion on why stochastic forward model does not yield a clear improvement over the deterministic model. This may be due to the limitation of the dataset or the prediction horizon which seems to be 2 second.”
>
>
> We have proposed a modification to the VAE posterior distribution for the stochastic model which now leads to a significant gain in performance over the deterministic model (please see top comment, and Section 2.1). Note also that we show, at least qualitatively, that the stochastic model without this modification does not respond very well to the input actions, even though it produces reasonable predictions. This is likely the reason for the suboptimal performance. The stochastic model with the modified posterior responds better, and also translates into better performance.
>
>
> >"The dataset is only 45 minutes which captured by a camera looking down a small section of the road. So the policies learned might only do lane following and occasionally doing collision avoidance. I would encourage the authors to look into more diverse dataset. See the paper DESIRE: Distant Future Prediction in Dynamic Scenes with Interacting Agents, CVPR 2017."
>
> Thank you for the pointer to this work. It seems very relevant and will be worth investigating in future work. We would like to note that two interesting features of our dataset are that it consists of real human driver behavior, and involves dense traffic. We believe this addresses an underexplored setting: as noted in the related work section, most other works deal with the problem of doing lane following or avoiding static obstacles in visually rich environments. Our setting instead focuses on visually simplified environments, but with complex and difficult to predict behavior by other drivers. The longer-term goal is to learn policies in visually rich settings with complicated driver behavior, and we believe solving this dataset is a step towards that goal. Also note that for autonomous driving, the success rate needs to be extremely high, and although our approach performs well in comparison to others, it is still far from 100%. We therefore believe that to obtain satisfactory performance, policies will have to learn fairly complex policies, and this dataset can serve as a useful testing environment.
>
> Please let us know if these address your concerns, and if you would consider updating your score if so.

---

### Official Review · AnonReviewer2 · 2018-11-05
**Review for "Model-Predictive Policy Learning with Uncertainty Regularization for Driving in Dense Traffic"**

**Rating:** 6
**Confidence:** 4

**Review:**

The paper addresses the difficulty of covariate shift in model-based reinforcement learning. Here, the distribution over trajectories during is significantly different for the behaviour or data-collecting policy and the target or optimised policy. As a mean to address this, the authors propose to add an uncertainty term to the cost, which is realised by the trace of the covariance of the outputs of a MC dropout forward model. The method is applied to driving in dense traffic, where even single wrong actions can be catastrophic.

I want to stress that the paper was a pleasure to read. It was extraordinarily straightfoward to follow, because the text was well aligned with the necessary equations.

The introduction and related work seem complete to me, with two exceptions:

- Depeweg, S., Hernandez-Lobato, J. M., Doshi-Velez, F., & Udluft, S.
  (2018, July). Decomposition of Uncertainty in Bayesian Deep Learning for Efficient and Risk-sensitive Learning. In *International Conference on Machine Learning* (pp. 1192-1201).
- Thomas, Philip S. *Safe reinforcement learning*. Diss. University of Massachusetts Libraries, 2015.

The work by Depeweg et al addresses quite the same question as the authors of this work, but with a broader scope (i.e. not limited to traffic) but very much the same machinery. There are some important theoretical insights in this work and the connection to this submission should be drawn. In particular, the proposed method needs to be either compared to this work or it needs to be clarified why it is not applicable.

The latter appears to be of less significance in this context, but I found robust offline policy evaluation underrepresented in the related work.

I wonder if there is a way for a neural network to "hack" the uncertainty cost. I suppose that the proposed approach is an approximation to some entropy term, and it would be informative to see how exactly.

The approach shown by Eq 1 appears to be an adhoc way of estimating whether the uncertainty resulting from an action is due to the data or the model. What happens if this approach is not taken?

The objective function of the forward model is only given in the appendix. I think it needs to be moved to the main text, especially because the sum-of-squares term indicates a homoskedastic Gaussian for a likelihood. This has implications for the uncertainty estimates (see point above).

Overall, the separation of data uncertainty/risk vs model uncertainty is not done. This indicates that heterskedastic environments are candidats where the method can fail, and this limitation needs to be discussed or pointed out.

Further, the authors did not observe a benefit from using a stochastic forward model. Especially, if the prior instead of the approximate posterior is used. My point would be that, depending on the exact grapical model and the way the sampling is done to train the policy, it is actually mathematically *right* to sample from the prior. This is also how it is described in the last equation of section 2.

## Summary

Overall, I liked the paper and the way it was written. However, there are some shortcomings, such as the comparison to the work by Depeweg et al, which does a very similar thing. Also, justifying the used heuristics as approximations to a principled quantity would help. It appears that the question why and how stochastic forward models should be used requires further investigation.

---

> ### Author Response · Authors · 2018-11-26
> **Thank you for the review (1/2)**
>
> Thank you for the constructive suggestions. We have made several updates to the paper based on them, and we provide answers to specific points below.
>
> >“The work by Depeweg et al addresses quite the same question as the authors of this work, but with a broader scope (i.e. not limited to traffic) but very much the same machinery. There are some important theoretical insights in this work and the connection to this submission should be drawn. In particular, the proposed method needs to be either compared to this work or it needs to be clarified why it is not applicable.”
>
>
> Thank you for pointing us to the work of Depeweg et al. [3]. It is indeed relevant and we have updated the paper to relate our work to theirs. The main difference between our approaches is that they use the framework of Bayesian neural networks trained with alpha-divergence minimization, whereas we use variational autoencoders trained with Dropout Variational Inference (VI).
>
> Both approaches aim to model aleatoric and epistemic uncertainties, but do so in different ways. Alpha-BNNs place a factorized Gaussian prior both over latent variables and network weights, and learn the parameters of these distributions by minimizing an energy function whose minimizer corresponds to a local minimum of alpha-divergences.
> Variational Autoencoders also represent latent variables as factorized Gaussians, whereas Dropout VI corresponds to placing a prior over network weights - specifically, a mixture of two Gaussians with small variances, with the mean of one component fixed at zero. As described in the new Section 2.3 which we have added, our approach corresponds to defining a variational distribution which is the composition of these two distributions.
>
> An advantage of using alpha-divergences over variational inference (pointed out in [1, 2, 3]) is that VI can underestimate model uncertainty by fitting to a local mode of the exact posterior, whereas alpha-divergence minimization can give better coverage of the distribution. However, there are also challenges associated with alpha-BNNs. One which was pointed out by [2] is that they require significant changes in existing deep learning models and code bases, and the functions they optimize are less intuitively interpretable by non-experts. We investigated the approach described in [2], which proposes a dropout-based reparameterization of the alpha-divergence objective, which seems to offer a balance between compatibility with existing frameworks and better-calibrated uncertainty estimates. However, this requires performing several stochastic passes through the forward model at training time in order to calculate the proposed loss. In our setup, doing 10 stochastic passes (the number used in the paper) required reducing the minibatch size from 64 to 8 to fit in memory, which significantly slowed down training. We did not obtain any reasonable results after 5 days of training on GPU, whereas with our current approach the model finishes training after 4 days. Since the minibatch size with the dropout-based alpha-divergence objective is 8x smaller than our original minibatch size, a rough estimate would place training time for the forward model at around 30 days. We note that the work of Depeweg et al. is applied to much lower-dimensional problems (2-30 dimensions, <100,000 transitions), whereas our setting involves high-dimensional images and a larger dataset (around 2 million transitions).  We believe that investigating alternate methods for uncertainty estimation in our setting would be interesting, but to do so thoroughly is best left for future work.
>
> References:
> [1] “Learning and Policy Search In Stochastic Dynamical Systems with Bayesian Neural Networks”, Depeweg S, Hernandez-Lobato H, Doshi-Velez F, Udluft S. ICLR 2017.
> [2] “Dropout Inference in Bayesian Neural Networks with Alpha-Divergences”, Yingzhen Li and Yarin Gal. ICML 2017.
> [3] “Decomposition of Uncertainty in Bayesian Deep Learning for Efficient and Risk-Sensitive Learning” Depeweg et al, ICML 2018.

---

> > ### Author Response · Authors · 2018-11-26
> > **Thank you for the review (2/2)**
> >
> >
> > >“I wonder if there is a way for a neural network to "hack" the uncertainty cost. I suppose that the proposed approach is an approximation to some entropy term, and it would be informative to see how exactly.”
> > “Overall, the separation of data uncertainty/risk vs model uncertainty is not done. This indicates that heterskedastic environments are candidats where the method can fail, and this limitation needs to be discussed or pointed out.”
> >
> >
> > In Section 2.3 we perform a similar uncertainty decomposition as Depeweg et. al (for covariance matrices, rather than scalar variances), and show that the uncertainty cost is obtained using the trace of the covariance matrix reflecting the epistemic uncertainty. Note also that the covariance matrix corresponding to the aleatoric uncertainty (second term in Equation 2) will change depending on the inputs. This allows our approach to handle heteroscedastic environments, where the aleatoric uncertainty will vary for different inputs. Intuitively, the latent variables in the VAE capture aleatoric uncertainty, whereas the change across different dropout masks reflects epistemic uncertainty.
> >
> > >”The objective function of the forward model is only given in the appendix. I think it needs to be moved to the main text, especially because the sum-of-squares term indicates a homoskedastic Gaussian for a likelihood. This has implications for the uncertainty estimates (see point above).”
> > >“Further, the authors did not observe a benefit from using a stochastic forward model. Especially, if the prior instead of the approximate posterior is used. My point would be that, depending on the exact grapical model and the way the sampling is done to train the policy, it is actually mathematically *right* to sample from the prior. This is also how it is described in the last equation of section 2.”
> >
> > We have moved the objective function to the main text. We have also proposed a modification to the VAE posterior distribution which now leads to a significant gain in performance of the stochastic model over the deterministic model, which is described in Section 2.1. (please also see top comment).
> >
> > Please let us know if these address your concerns, and if you would consider updating your score if so.

---

### Official Review · AnonReviewer1 · 2018-11-06
**An Ok paper that combines  dropout methods with learning policy using observational data.**

**Rating:** 6
**Confidence:** 5

**Review:**

- Does the paper present substantively new ideas or explore an under explored or highly novel question?

Somewhat, the paper combines two popular existing approaches (Imitation Learning,  Model Based Control and Uncertainty Quantification using Dropout).  The novelty is in  combining pre-existing ideas.

- Does the results substantively advance the state of the art?

No, the compared methods are not state-of-the-art.

- Will a substantial fraction of the ICLR attendees be interested in reading this paper?

Yes. I think that the topics of this paper would be very interesting to ICLR attendees.

-Quality:

Unclear motivation to penalize prediction uncertainty to make the predicted states stay in the training data.  Also, in some cases references to existing work that includes real robotic systems is out of context at minimum. So yes there are similarities between this paper and existing works  on  learning control for robotics systems using imitation learning, model based control and uncertainty aware cost function. However there is a profound difference in terms of working in simulation and working with a real system for which model and environment uncertainty is a very big issue. There are different challenges in working with a real uncertain system which you will have to actuate,  and working with set of images for making predictions in simulation.



-Clarity:

Easy to read. Experimental evaluation is clearly presented.

-Originality:

Similar uncertainty penalty was used in other paper (Kahn et al. 2017).  Therefore the originality is in some sense reduced.

- Would I send this paper to one of my colleagues to read?

Yes I would definitely send this paper to my colleagues.

- General Comment:

Dropout can be used to represent the uncertainty/covariance of the neural network model. The epistemic uncertainty, coming from the lack of data, can be gained through Monte Carlo sampling of the dropout-masked model during prediction. However, this type of uncertainty can only decrease by adding more explored data to current data set. Without any addition of data, the  variance reduction, which results  by penalizing the high variance during training, might indicate over-fitting to the current training data. As the penalty forces the model to predict states only in the training dataset, it is unclear how this shows better test-time performance. The output of the policy network will simply be biased towards the training set as a result of the uncertainty cost. More theoretical explanation is needed or perhaps some intuition.

This observation is also related to the fact that the model based controller used  is essentially a  risk sensitive controller.

---

> ### Author Response · Authors · 2018-11-26
> **Thank you for the review**
>
> Thank you for the helpful suggestions, we have updated the paper. Please see our answers to specific points below:
>
> >“Unclear motivation to penalize prediction uncertainty to make the predicted states stay in the training data”
> “More theoretical explanation is needed or perhaps some intuition.”
>
> As requested, we have added a section (Section 2.3 and Appendix B), where we show that our approach can be seen as training a Bayesian neural net with latent variables using variational inference. We also perform a similar uncertainty decomposition as Depeweg et. al [1], and show that the uncertainty cost is obtained using the trace of the covariance matrix reflecting the epistemic uncertainty.
>
> >“Without any addition of data, the  variance reduction, which results  by penalizing the high variance during training, might indicate over-fitting to the current training data. As the penalty forces the model to predict states only in the training dataset, it is unclear how this shows better test-time performance. The output of the policy network will simply be biased towards the training set as a result of the uncertainty cost.
>
> We would like to clarify that the uncertainty penalty does not necessarily bias the policy network towards the training trajectories, but rather toward the states where the forward model has low uncertainty. This includes the training trajectories, but it also includes regions of the state space where the forward model generalizes well, which were not seen during training. The prediction results, which are obtained by feeding initial states from the testing set which the forward model was not trained on, still look reasonable, which indicates that the forward model is able to generalize fairly well. Note also that we evaluate the trained policy network on trajectories from the testing set, which the forward model was not trained on.
>
> >”Also, in some cases references to existing work that includes real robotic systems is out of context at minimum. So yes there are similarities between this paper and existing works  on  learning control for robotics systems using imitation learning, model based control and uncertainty aware cost function. However there is a profound difference in terms of working in simulation and working with a real system for which model and environment uncertainty is a very big issue. There are different challenges in working with a real uncertain system which you will have to actuate,  and working with set of images for making predictions in simulation.”
>
>
> We agree that there is a big difference between our setup and a real robotic system. We felt it fair to include references to other work in imitation learning and model-based control, even if the setups are quite different. We are happy to update our related work section with additional references, if you have any suggestions.
>
> Please let us know if these address your concerns, and if you would consider updating your score if so.
>
> [1] “Decomposition of Uncertainty in Bayesian Deep Learning for Efficient and Risk-Sensitive Learning” Depeweg et al, ICML 2018.

---

### Author Response · Authors · 2018-11-26
**Updated Paper**

We would like to thank all the reviewers for their helpful feedback. We have made several updates to the paper which we hope address the reviewers’ concerns, which we describe below. We give more detailed responses to the individual comments.

Both Reviewer 2 and Reviewer 3 mentioned the fact that the stochastic model did not yield an improvement over the deterministic model as a limitation. In the updated version of the paper we propose a modified posterior distribution for the VAE, which gives improved performance relative to both the standard stochastic model and the deterministic model. This modification is simple to implement, and involves sampling the latent variable from the prior, rather than posterior, a fraction of the time during training. In addition to improving the performance of the trained policies (in terms of success and distance travelled), upon visual inspection (shown at the URL) this modification makes the forward model more responsive to the input actions, which we believe is the reason for the standard stochastic model’s suboptimal performance. This modification can be seen as “dropping out” the latent code with some probability, and although simple, we are not aware of it being proposed elsewhere in the literature.


Both Reviewer 1 and Reviewer 2 mentioned they would like to see more theoretical explanation. We have added a new section (Section 2.3 and Appendix B) which shows that our approach can be viewed as training a Bayesian neural network with latent variables using variational inference. We show that the loss function which we optimize is in fact an approximation to the negative evidence lower bound obtained by using a variational distribution which is the composition of a diagonal Gaussian (over latent variables) and the dropout approximating distribution (over model parameters) described in [1]. We also perform a decomposition of the covariance of the distribution over predictions induced by this approximate posterior (similar to [2]) into two covariance matrices, which represent the aleatoric and epistemic uncertainties.  Our uncertainty penalty is in fact penalizing the trace of the matrix representing the epistemic uncertainty.

We have moved certain parts of the main text to the appendix to make room for this new section and stay within the page limit. We have also rerun the experiments with different seeds to obtain more robust performance estimates, and made some changes in our training procedure/hyperparameters (these are detailed in the Appendix, and will be available in our code release). Note that the MPUR results are now somewhat higher than in the first version, although their relative performance is similar (i.e, deterministic and stochastic are still similar to each other, although the stochastic model with our modified posterior is better than both).

[1]: "Dropout as a Bayesian Approximation: Representing Model Uncertainty in Deep Learning", Gal and Ghahramani. ICML 2016.

[2]: “Decomposition of Uncertainty in Bayesian Deep Learning for Efficient and Risk-Sensitive Learning” Depeweg et al, ICML 2018.

---

### Author Response · Authors · 2018-11-26
**Additional Updates.**

We have made a few additional formatting changes, please see the updated version.

---

### Meta-Review · Area_Chair1 · 2018-12-14

**Confidence:** 4
**Recommendation:** Accept (Poster)

**Metareview:**

Reviewers are in a consensus and recommended to accept after engaging with the authors. Please take reviewers' comments into consideration to improve your submission for the camera ready.